# Cognition and Behavior of COVID-19 Vaccination Based on the Health Belief Model: A Cross-Sectional Study

**DOI:** 10.3390/vaccines10040544

**Published:** 2022-04-01

**Authors:** Zemin Cai, Wei Hu, Shukai Zheng, Xilin Wen, Kusheng Wu

**Affiliations:** 1Department of Preventive Medicine, Shantou University Medical College, Shantou 515041, China; 19zmcai@stu.edu.cn (Z.C.); 15skzheng@stu.edu.cn (S.Z.); 20xlwen@stu.edu.cn (X.W.); 2Department of Epidemiology, School of Public Health, Sun Yat-sen University, Guangzhou 510080, China; huwei59@mail2.sysu.edu.cn

**Keywords:** cognition, behavior, COVID-19 vaccination, health belief model

## Abstract

Background: Vaccination is the most effective method for the prevention of COVID-19. However, willingness to be vaccinated is not consistent. This study aimed to explore vaccine cognition, risk perception, and health behavior of COVID-19 in China. Methods: A cross-sectional survey was performed in Guangdong province, China, including demographic characteristics, health status and preventive behaviors, cognition of COVID-19 vaccination, and the health belief model (HBM). Results: A total of 1640 participants were recruited. The main access to information about COVID-19 and vaccination as through official news and broadcasts (67.3%), social network software (58.7%), and professional popularization (46.2%). The precautions taken were wearing a mask (67.0%) and avoiding gathering together (71.3%). COVID-19 vaccination acceptability was different among different age groups and educational levels (*p* < 0.001). The major reasons for accepting vaccination included that it was an effective way to prevent COVID-19 (61.8%) and that it was required by working units/schools (51.1%). The fitting effect indexes of the (HBM) Model 2 showed better fitting than those of Model 1. In Model 2, perceived benefits (OR = 3.13, 95% CI: 1.79–5.47), cues to action (OR = 2.23, 95% CI: 1.60–3.11), and different occupations (OR = 1.13, 95% CI: 1.04–1.23) were positively correlated with vaccine acceptance; while perceived susceptibility (OR = 0.47, 95% CI: 0.30–0.74) and perceived barriers (OR = 0.44, 95% CI: 0.29–0.69) were negative factors associated with vaccine acceptance. Conclusion: Different sociodemographic characteristics lead to differences in acceptance of vaccination, and the publicity and credibility of government play an indispensable role in epidemic control. The establishment of the HBM further predicted that perceived susceptibility to COVID-19, benefits of vaccination, barriers of cognition, and cue to action were the influencing factors of intention and health behaviors.

## 1. Introduction

In December 2019, the first patient of coronavirus disease 2019 (COVID-19) was reported in Hubei Province, China [1]. Up to 28 January 2022, about 138,050 people have been cumulatively diagnosed in China, and 5700 people had died [2]. Nearly 200 countries worldwide have confirmed COVID-19 cases, and more than 360 million people have been diagnosed [3]. As for the development of the global pandemic, human beings will face the severe situation of its strong infectivity and great harm for a long time [4]. The promotion of COVID-19 vaccine research and application is an important means of epidemic prevention.

At present, the COVID-19 vaccine has been widely used all over the world, and the cumulative quantity of the Chinese COVID-19 vaccine doses has exceeded 2990 million. Its safety and effectiveness have been verified to a certain extent [5]. However, studies showed that people’s willingness to be vaccinated is not consistent, and skepticism and hesitancy toward getting vaccinated are universal [6]. COVID-19 vaccination rates are low among the general population and medical workers in many countries. Geofford and Philipson [7] point out that simply subsidizing vaccine costs and having mandatory vaccinations will not eliminate the disease or increase the number of vaccinations. Only increasing the demand for vaccines can eliminate the disease. Therefore, it is necessary to increase people’s active demand for the COVID-19 vaccine through scientific research.

However, an active demand for vaccines is not uniform among different populations, because heterogeneity among individuals with different socio-economic characteristics, health status, and personal experiences can cause significant discrepancies in their willingness to be vaccinated [8,9]. Besides, risk perception is an important influence on health behaviors such as vaccination [10,11]. When people have the wrong risk perception of an event, it can have a significant impact on their willingness to adopt healthy behaviors. When faced with public health emergencies, risk perception is an important factor affecting individual behavior. During the COVID-19 pandemic, it is of great significance to understand the characteristics of risk perception, so as to explore the heterogeneity of vaccination cognition and attitudes among different people.

Kasl and Cobb [12] first put forward the concept of health behavior, which is that behavior performed by a person who believes he or she is healthy in order to prevent disease or to diagnose the disease early in the asymptomatic phase. Vaccination of an uninfected person with the COVID-19 vaccine may be considered as a health behavior, for the reason that risk perception is at the core of health behavior theory [13]. Therefore, health research on risk perception based on the health behavior theory is needed. The health belief model (HBM) is an indispensable theory in medicine, guiding work in the health field. It founded and developed by scholars such as Rosenstock [14,15]. The HBM explains why people adopt certain healthy behaviors (or quit certain unhealthy behaviors) on the basis of taking social psychological factors into account.

From the perspective of the basic elements of the HBM and the application of empirical research, demographic characteristics, sociopsychological variables, social influences, perceived benefits, costs, perceived risks, and other factors are all important for the adoption of healthy behaviors. In the current COVID-19 pandemic, there are few reports on the study of COVID-19 vaccination intention using the HBM theory. This study used the HBM to explore COVID-19 vaccine cognitions, risk perceptions, and health behavior (vaccination) in the Chinese population.

## 2. Materials and Methods

### 2.1. Study Design and Participants

The cross-sectional study was carried out in Guangdong Province, China, from 25 June to 25 December 2021. The sample size was calculated in accordance with the formula: N = [Max (dimensions) × 10] × [1 + 20%]. The research was approved by the human ethics committee of Shantou University Medical College in terms of the Declaration of Helsinki (authorization number: SUMC-2021-90). The participants provided informed consent prior to the survey.

Wenjuanxing (https://www.wjx.cn/ (accessed on 26 August 2019)) is a professional online survey platform. After designing the questionnaire and setting properties, the questionnaire was distributed to the participants through Wechat, QQ and other ways using a QR code or website link. Epidemiologists and psychologists were invited to fill in the questionnaire independently, and gave feedback on the rationality and wording of the first draft of the questionnaire. According to the opinions, the final form of a formal questionnaire was modified, and a large-scale convenience sampling survey was carried out. Each IP address was allowed to fill in only once.

### 2.2. Questionnaires

The survey consisted of four sections: (1) sociodemographic characteristics of the participants, including gender, age, occupation, educational level, marital status, residence, and household income; (2) health status and preventive behaviors, including health status, contact history of COVID-19 in previous 14 days, access to information, and precautions; (3) cognition of COVID-19 vaccination, including willingness for COVID-19 vaccination, having had the COVID-19 vaccination or not, cognition of the effectiveness of the COVID-19 vaccination, cognition of different vaccine producers, cognition of the effective time, and reasons for accepting or not accepting COVID-19 vaccination; and (4) the HBM included perceived susceptibility, perceived severity, perceived benefits, perceived barriers, cues to action, and health behavior. On the scale of the HBM, each dimension has three to four items, and the score of each item ranged from 1 (completely disagree) to 5 (agree completely) (Appendix A). The general conceptual framework of the HBM is shown in Figure 1.

### 2.3. Statistical Analysis

Statistical analyses were implemented by IBM SPSS Statistics 26.0 (IBM Corp., Armonk, NY, USA) and R 4.1.0 (R Foundation for Statistical Computing; https://www.r-project.org, accessed on 26 August 2019). GraphPad Prism 9.0 (GraphPad Software Inc., San Diego, CA, USA) was applied to graphing. Descriptive analyses were used for demographic data, shown as a percentage. The independent sample t-tests and one-way ANOVA were used for the continuous variables. The chi-squared test was used to analyze the categorical variables. The region classification (low-risk, medium-risk, or high-risk) among demographic characteristics, willingness to be vaccinated, and cognitive level of COVID-19 among different age groups and educational levels were performed by chi-squared tests. The willingness to be vaccinated and cognitive level of COVID-19 were also performed by independent sample t-tests and ANOVA. All tests were two-sided and *p* < 0.05 was considered to be statistically significant. The HBM was analyzed by binary logistic regression. The Rho-square (R^2^), final logarithmic likelihood ratio (LLR), Akaike information criterion (AIC), and Bayesian information criterion (BIC) were used to estimate the fitting results of statistical models. R^2^, which is the percentage of variation in the predictive variable that explains the resulting variable, ranging from 0 to 1, is the square of the correlation coefficient between the observed actual results and the predicted values from the model. The higher the R^2^, the better. The basis of LLR is to make the most likely outcome of a test as the best estimate when the test results are available, so the bigger the LLR, the better. The basis of AIC is to punish the behavior with extra variables in the model. Every time a new variable is added, it adds a penalty value to control the extra predictive variables. The lower the AIC, the better. BIC is another variant of AIC, which makes use of the Bayes principle. When new variables are added to the model, BIC will be punished more severely than AIC.

## 3. Results

### 3.1. Demographic Characteristics

A total of 1712 participants were recruited in this population-based survey in Guangdong, the most developed province in China. After screening the validity and finished time, 1640 questionnaires were retained as valid, including 733 males and 907 females. Most of the participants ranged from 20 to 29 (30.9%) or ≥40 (31.5%) years old. The occupations were included medical workers (18.2%), students (18.6%), company employees (12.3%), and social workers (7.3%). The majority of the participants had a high educational level, including undergraduate/junior college (56.8%) or postgraduate or above (20%). Among them, 58.9% were married, and 84.7% lived in urban area. The region classification of risk was defined for the latest month according to COVID-19 cases reported. Most participants resided in the low-risk areas (90.9%) (Figure 2), with a statistical significance of region classification among different ages and marital statuses (*p* < 0.001) (Table 1).

### 3.2. Health Status and Preventive Behaviors

Most (67.1%) of the participants were in health status, while 26.5% had a chronic disease, 3.0% had an infectious disease, and 2.9% had catarrh symptoms, such as fever and headache/fatigue (Figure 3A). Due to China’s epidemic prevention policy, 93.7% did not have an infection history of COVID-19, while 2.9% had had recent trips to high- or medium-risk areas (Figure 3B).

The main access to COVID-19 cognitive and vaccination information was though official news and broadcasts (67.3%), social software (58.7%), timely messages from software (42.4%), internet searching (42.1%), and popularization of science from professionals (46.2%) (Figure 4A). The precautions the participants took were wearing a mask (67.0%), washing hands and disinfecting furniture (71.7%), avoiding gathering together (71.3%), exercising (55.6%), keeping healthy living habits (48.8%), and a balanced diet (37.3%) (Figure 4B).

### 3.3. Willingness to Be Vaccinated and Cognitive Level of COVID-19

Willingness to be vaccinated was different in different age groups (*p* < 0.001). COVID-19 vaccination acceptability was also different among age and educational levels (all *p* < 0.001), as was the cognition of different vaccine producers’ effectiveness, and the effective time (all *p* < 0.001) (Table 2).

The reasons for accepting the COVID-19 vaccination were as follows: being afraid of infection with COVID-19 (44.9%), ones’ profession has a risk of infection with COVID-19 (50.9%), believing that vaccines are an effective way to prevent disease (61.8%), COVID-19 vaccination is required by working units/schools (51.1%), business trips (19.7%), and most people accept COVID-19 vaccination (23.3%). The reasons for not accepting the COVID-19 vaccination were as follows: having no time right now (17.2%), having contraindications to vaccination (31.0%), it is hard to book the COVID-19 vaccination (20.7%), and concerns about the impact of vaccination on the fetus/baby during pregnancy, pregnancy, or lactation (17.2%) (Figure 5).

### 3.4. HBM-Based Heterogeneity Factors

In binary logistic regression analysis without the demographic factors and health factors (Model 1), perceived benefits (OR = 2.76, 95% CI: 1.62–4.70, *p* < 0.001) and cues to action (OR = 2.30, 95% CI: 1.71–3.09, *p* < 0.001) were positive correlations to vaccine acceptance; perceived susceptibility (OR = 0.48, 95% CI: 0.31–1.57, *p* = 0.001) and perceived barriers (OR = 0.49, 95% CI: 0.32–0.76, *p* = 0.001) were negative correlations to vaccine acceptance. Incorporating all covariates into the analysis (Model 2), perceived benefits (OR = 3.13, 95% CI: 1.79–5.47, *p* < 0.001), cues to action (OR = 2.23, 95% CI: 1.60–3.11, *p* < 0.001), and different occupations (OR = 1.13, 95% CI: 1.04–1.23, *p* = 0.003) were positively correlated with vaccine acceptance; perceived susceptibility (OR = 0.47, 95% CI: 0.30–0.74, *p* = 0.001) and perceived barriers (OR = 0.44, 95% CI: 0.29–0.69, *p* < 0.001) were negative factors associated with vaccine acceptance (Table 3).

During the modeling process, all the key variables were preserved. When the other variables were introduced, the stepwise regression was adopted, and the variables were introduced into Model 1 according to category and degree of importance. When each new explanatory variable was introduced, indices to measure the fitting effect of statistical models such as the R^2^, AIC, BIC, and the final LLR were used to judge whether the newly introduced explanatory variables could help the model to fit better. The R^2^, LLR, AIC, and BIC of Model 1 were 0.16, −149.13, 467.26, and 953.26, respectively. The model estimations of Model 2 were 0.18, −143.63, 460.26, and 897.86, respectively, which shows that Model 2 have a better fitting effect than Model 1 (Table 3).

## 4. Discussion

This is the first investigation into cognition and health behavior toward COVID-19 vaccination based on the HBM theory during the COVID-19 pandemic. This study was based on a network platform and had a high response rate and the questionnaire was examined by epidemiologists and psychologists. According to our study, 90.5% of the participants would have liked to accept the COVID-19 vaccination, compared to an acceptance of 86.2% among the whole population in China [16]. This study showed that young adults and people with higher educational levels had higher acceptance and cognition of COVID-19. We explored the correlation of the basic HBM to vaccinate intention and behavior, and also analyzed the associations with demographic and health factors, concluding that perceived benefits, cue to action, and occupation were significantly associated with higher vaccine acceptance, while perceived susceptibility and perceived barriers of COVID-19 were negative factors for vaccine acceptance.

The participants were mainly distributed in low-risk regions (90.9%), and the region classifications among age and marital status were different (*p* < 0.001). The reason for this phenomenon is that the majority of participants were more than 20 years old. This age group, which includes university students and workers from all walks of life, is also the most active in social activities, reflecting the impact of the pandemic on the population, especially the main economic forces [17]. In 2021, six cities (Guangzhou, Shenzhen, Dongguan, Zhuhai, Foshan, and Zhanjiang) of Guangdong province were classified into medium-risk or high-risk regions, and most regions of Guangdong province were low-risk regions (Figure 2). However, the different demographic characteristics meant that people held different cognitions of COVID-19 vaccination. Married residents paid more attention to vaccine knowledge for the consideration of their families, and people with different incomes and occupations took different attitudes towards their own health and the accessing of information [18], which was consistent with our study.

The government is still an important factor influencing the cognition, attitude, and practice of COVID-19 vaccination. The acceptance of COVID-19 vaccination by the participants was 90.9%, which is higher than that of America or Denmark (57.6–73.9%) [5,19]. In addition, precautions become widespread among the population, including wearing a mask, washing hands and disinfecting furniture, avoiding gathering together, reflecting the prevention effect. The government’s attitude and policies towards the COVID-19 vaccination affected the acceptance rate to a large extent. The government set up publicity for the high-risk population and combined the efforts of various sectors of society, using official news and social software to convey information about vaccine development to the public, so as to verify and correct the negative reports and exaggerated propaganda and strengthen residents’ cognition of COVID-19 vaccination.

The majority of HBM dimensions were associated with COVID-19 vaccine acceptance. In this study, participants who perceived the severity of COVID-19, the benefits of vaccination, and cues to action were more likely to accept the COVID-19 vaccination. On the contrary, perceived barriers and susceptibility were negatively associated with vaccine acceptance. This is consistent with the study of Martin [20]. In the study by Wong et al. [21], the public also attached great importance to the efficacy and safety of the vaccination. In the present study, the perceived severity of COVID-19 infection had no statistical significance with vaccine acceptance, reflecting the fact that COVID-19 is regarded as a mild disease only if the person has been in risk situations. This means that the dissemination of information and education about the COVID-19 vaccine may be different from that of other vaccines whose primary purpose is to prevent infection. Rather, we should attach great significance to perceived susceptibility to COVID-19. The association between cue to action and vaccine acceptance demonstrates that cue to action is an important dimension of the HBM, with publicity from the government being the crucial cue. These studies showed that the dimensions of the HBM can explain health behavior (vaccine acceptance), as in [22], which illustrated the importance of the HBM dimensions in predicting behaviors related to influenza vaccination.

### Limitations and Strengths

This study had some limitations. Firstly, it relied on network investigation due to the epidemic situation, and a field epidemiological investigation and strict stratified sampling design might be better. Secondly, it just reflected the current status of COVID-19 vaccine cognition and acceptance, and a longitudinal study might help research the variations in cognition and acceptance of COVID-19 vaccination. Despite these limitations, the strengths of the study support this innovative research. We considered demographic and health factors in the acceptance of vaccination, bringing a series of reasons into consideration. In addition, a HBM was established to predict the association of demographic factors, health factors, HBM dimensions, and health behaviors. Our work will not only provide a reference for the research of vaccination in the situation of infectious diseases, but also promote the application of health behavior theory in this kind of situation. In the future, protection motivation and a planned behavior model can be considered in the study, so as to further define the influence of COVID-19 vaccination, and provide a reference for the research of vaccination in the situation of infectious diseases.

## 5. Conclusions

This study examined the demographic factors, health factors, and behavior predictors for intention and behavior related to COVID-19 vaccination, based on the HBM. Our study highlights that people with different social and demographic characteristics have a different acceptance of vaccination, and the publicity and credibility of government play an indispensable role in epidemic prevention and control. The establishment of the HBM further predicts that perceived susceptibility to COVID-19, benefits of vaccination, barriers of cognition, and cue to action are the affecting factors of intention and health behaviors.

## Figures and Tables

**Figure 1 vaccines-10-00544-f001:**
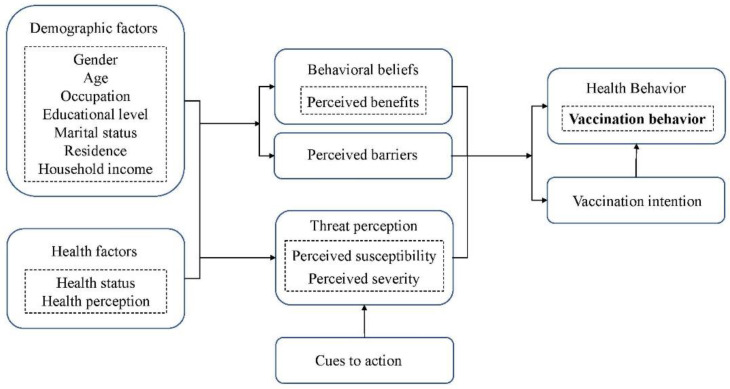
Health belief model.

**Figure 2 vaccines-10-00544-f002:**
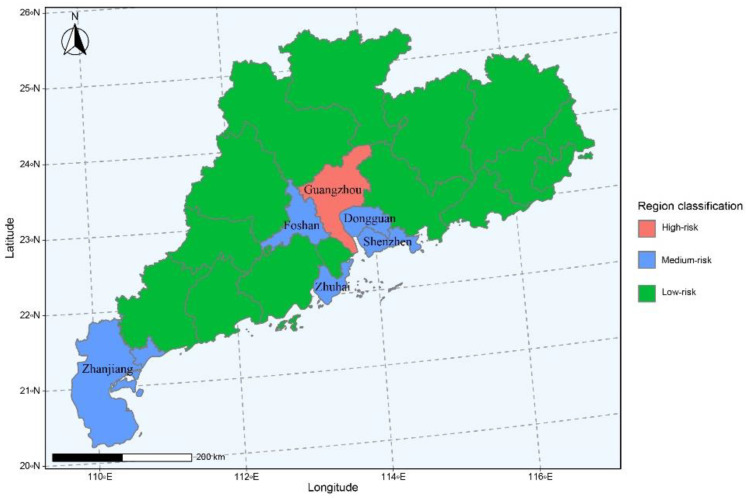
Region classification of COVID-19 epidemic in Guangdong province, China. There are 21 prefecture-level cities in Guangdong Province. During 2021, Guangzhou was categorized as a high-risk area. The medium-risk areas included Shenzhen, Dongguan, Foshan, Zhuhai, and Zhanjiang. The rest of cities were low-risk.

**Figure 3 vaccines-10-00544-f003:**
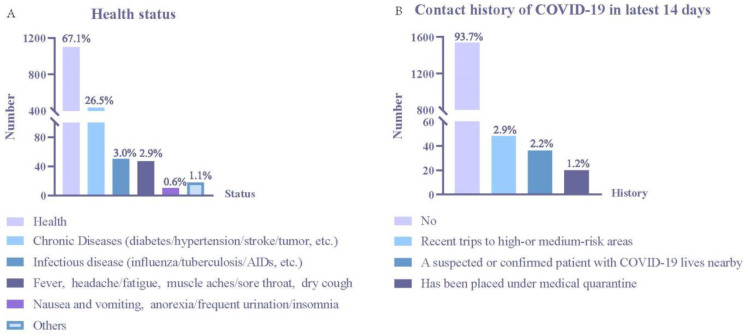
Health status and contact history of COVID-19. (**A**) Health status. (**B**) Contact history of COVID-19 in latest 14 days.

**Figure 4 vaccines-10-00544-f004:**
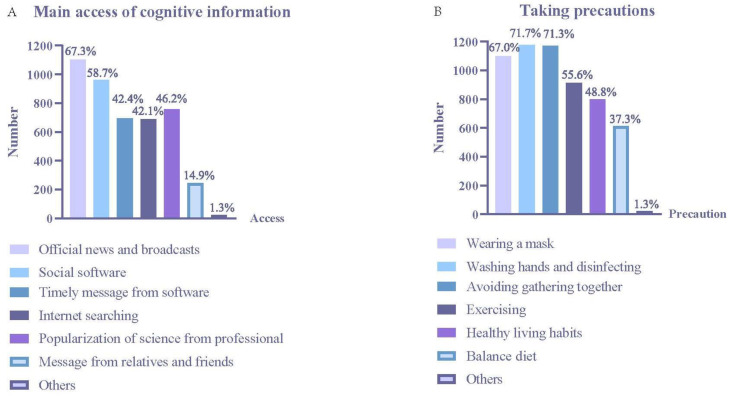
Access to information and preventive measures. (**A**) Main access of cognitive information. (**B**) Taking precautions.

**Figure 5 vaccines-10-00544-f005:**
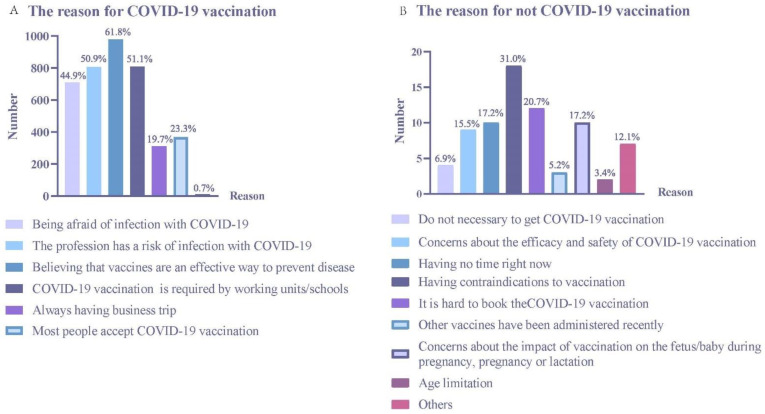
Reasons for accepting or not accepting COVID-19 vaccination. (**A**) The reason for COVID-19 vaccination. (**B**) The reason for not COVID-19 vaccination.

**Table 1 vaccines-10-00544-t001:** Demographic characteristics of the participants.

Variables	Total	Region Classification			*χ* * ^2^ *	*p*-Value
		Low-Risk	Medium-Risk	High-Risk		
Gender						
Male	733 (44.7)	673 (41.0)	29 (1.8)	31 (1.9)	1.51	0.47
Female	907 (55.3)	819 (49.9)	38 (2.3)	50 (3.0)		
Age						
≤19	115 (7.0)	90 (5.5)	14 (0.8)	11 (0.7)	24.73	<0.001
20–29	539 (32.9)	478 (29.1)	27 (1.7)	34 (2.1)		
30–39	437 (26.6)	404 (24.6)	17 (1.0)	16 (1.0)		
≥40	549 (33.5)	516 (31.5)	13 (0.8)	20 (1.2)		
Occupation						
Medical workers	299 (18.2)	267 (16.3)	10 (0.6)	22 (1.3)	38.46	0.06
Scientific researchers	61 (3.7)	56 (3.4)	1 (0.1)	4 (0.2)		
Health system staff	72 (4.4)	66 (4.0)	4 (0.2)	2 (0.1)		
Civil servants/institution staff	87 (5.3)	81 (4.9)	3 (0.2)	3 (0.2)		
Teaching staff	97 (5.9)	90 (5.5)	5 (0.3)	2 (0.1)		
Students	304 (18.6)	258 (15.8)	18 (1.1)	28 (1.7)		
Company employees	202 (12.3)	186 (11.3)	9 (0.5)	7 (0.4)		
Social workers	120 (7.3)	114 (7.0)	3 (0.2)	3 (0.2)		
Service staff	108 (6.6)	103 (6.3)	4 (0.2)	1 (0.1)		
Farmers/workers	39 (2.4)	38 (2.3)	0 (0.0)	1 (0.1)		
Liberal professions	57 (3.5)	51 (3.1)	4 (0.2)	2 (0.1)		
Retirees	29 (1.8)	29 (1.8)	0 (0.0)	0 (0.0)		
Others	165 (10.1)	153 (9.3)	6 (0.4)	6 (0.4)		
Educational level						
Junior high school or below	128 (7.8)	122 (7.4)	4 (0.2)	2 (0.1)	7.70	0.26
Senior high school/technical secondary school	253 (15.4)	237 (14.5)	8 (0.5)	8 (0.5)		
Undergraduate/junior college	931 (56.8)	836 (51.0)	41 (2.5)	54 (3.3)		
Postgraduate or above	328 (20.0)	297 (18.1)	14 (0.9)	17 (1.0)		
Marital status						
Spinsterhood	495 (30.2)	417 (25.4)	37 (2.3)	41 (2.5)	44.19	<0.001
Married	966 (58.9)	905 (55.2)	23 (1.4)	38 (2.3)		
Divorced	152 (9.3)	143 (8.7)	7 (0.4)	2 (0.1)		
Widowed	27 (1.6)	27 (1.6)	0 (0.0)	0 (0.0)		
Residence						
Urban area	1389 (84.7)	1259 (76.8)	62 (3.8)	68 (4.1)	3.32	0.19
Rural area	251 (15.3)	233 (14.2)	5 (0.3)	13 (0.8)		
Monthly household income (RMB)						
≤3000	273 (16.6)	256 (15.6)	9 (0.5)	8 (0.5)	9.16	0.17
3001–5000	385 (23.5)	359 (21.9)	11 (0.7)	15 (0.9)		
5001–10,000	479 (29.2)	426 (26.0)	25 (1.5)	28 (1.7)		
>10,000	503 (30.7)	451 (27.5)	22 (1.3)	30 (1.8)		

The region’s classification of risk was defined for the latest month. Low-risk area: no confirmed COVID-19 cases or no additional confirmed COVID-19 cases within 14 days; medium-risk area: additional cases confirmed within 14 days, and the total number of COVID-19 cases was no more than 50, or, a total of more than 50 COVID-19 cases were confirmed, and no aggregated outbreak had occurred within 14 days; high-risk area: a total of more than 50 COVID-19 cases were confirmed, and aggregated outbreak occurred within 14 days.

**Table 2 vaccines-10-00544-t002:** Willingness to be vaccinated and cognitive level of COVID-19.

Variables	Age				Educational Level ^#^			
	≤19	20–29	30–39	≥40	I	II	III	IV
Are you willing to have the COVID-19 vaccination?								
Yes	110 (6.7)	533 (32.6)	436 (26.5)	542 (33.1)	128 (7.8)	248 (15.1)	920 (56.1)	325 (19.8)
No	5 (0.3)	6 (0.3)	1 (0.1)	7 (0.4)	0 (0.0)	5 (0.3)	11 (0.7)	3 (0.2)
*p*-value	0.003				0.37			
Have you had the COVID-19 vaccination?								
Yes, all the injections have been administered	92 (5.6)	465 (28.4)	406 (24.7)	522 (31.8)	120 (7.3)	237 (14.4)	843 (51.4)	285 (17.4)
Yes, but not all of the vaccinations have been administered	18 (1.1)	51 (3.1)	17 (1.0)	11 (0.7)	5 (0.3)	11 (0.7)	48 (3.0)	33 (2.0)
No	5 (0.3)	23 (1.4)	14 (0.9)	16 (1.0)	3 (0.2)	5 (0.3)	40 (2.4)	10 (0.6)
*p*-value	<0.001				<0.001			
Do you think vaccination is an effective way to prevent disease?								
Yes	109 (6.6)	500 (30.6)	417 (25.4)	525 (32.0)	123 (7.5)	248 (15.1)	877 (53.5)	304 (18.5)
No	1 (0.1)	3 (0.2)	0 (0.0)	1 (0.1)	0 (0.0)	1 (0.1)	2 (0.1)	1 (0.1)
Not sure	5 (0.3)	36 (2.2)	20 (1.2)	23 (1.4)	5 (0.3)	4 (0.2)	52 (3.2)	23 (1.4)
*p*-value	0.26				0.11			
Do you think the vaccinations from different producers will have an effect on the body?								
Yes	20 (1.2)	170 (10.3)	230 (14.1)	324 (19.8)	84 (5.2)	170 (10.4)	388 (23.7)	103 (6.3)
No	32 (2.0)	80 (4.9)	42 (2.5)	45 (2.7)	4 (0.2)	12 (0.7)	138 (8.4)	47 (2.9)
Not sure	63 (3.8)	289 (17.7)	165 (10.0)	180 (11.0)	40 (2.4)	71 (4.3)	405 (24.7)	178 (10.8)
*p*-value	<0.001				<0.001			
How long do you think the full dose of the COVID-19 vaccine will last?								
<6 months	21 (1.3)	147 (8.9)	130 (7.9)	178 (10.9)	42 (2.6)	99 (6.0)	255 (15.5)	81 (4.9)
≥6 months	36 (2.2)	188 (11.5)	215 (13.1)	264 (16.1)	63 (3.8)	123 (7.6)	395 (24.2)	124 (7.6)
Lifelong	3 (0.2)	14 (0.9)	18 (1.1)	23 (1.4)	9 (0.5)	9 (0.5)	33 (2.0)	6 (0.4)
Not sure	55 (3.3)	190 (11.6)	74 (4.5)	84 (5.1)	14 (0.9)	22 (1.3)	248 (15.1)	117 (7.1)
*p*-value	<0.001				<0.001			

^#^ Educational level: I, junior high school or below; II, senior high school/technical secondary school; III, undergraduate/junior college; IV, postgraduate or above.

**Table 3 vaccines-10-00544-t003:** Factors associated with acceptance of COVID-19 vaccine by binary logistic regression.

Variables	Model 1			Model 2		
	B	OR (95% CI)	*p*-Value	B	OR (95% CI)	*p*-Value
HBM						
Perceived severity	0.16	1.18 (0.88,1.57)	0.27	0.13	1.13 (0.84,1.54)	0.42
Perceived susceptibility	−0.73	0.48 (0.31,0.75)	0.001	−0.75	0.47 (0.30,0.74)	0.001
Perceived benefits	1.01	2.76 (1.62,4.70)	<0.001	1.14	3.13 (1.79,5.47)	<0.001
Perceived barriers	−0.72	0.49 (0.32,0.76)	0.001	−0.81	0.44 (0.29,0.69)	<0.001
Cues to action	0.83	2.30 (1.71,3.09)	<0.001	0.80	2.23 (1.60,3.11)	<0.001
Health behavior	−0.04	0.96 (0.74,1.24)	0.76	−0.03	0.97 (0.75,1.27)	0.84
Demographic factors						
Gender				0.46	1.58 (0.88,2.83)	0.13
Age				0.08	1.08 (0.60, 1.95)	0.80
Occupation				0.12	1.13 (1.04, 1.23)	0.003
Educational level				0.08	1.08 (0.71, 1.64)	0.71
Marital status				−0.34	0.71 (0.38, 1.35)	0.30
Residence				0.44	1.56 (0.74, 3.28)	0.24
Household income				0.25	1.28 (0.97, 1.69)	0.08
Health factors						
Health status				0.70	2.02 (0.86, 4.71)	0.10
Health perception				−0.64	0.53 (0.12,2.43)	0.41
**Comparison of the two models**						
Index of modelfitting effect			Model 1		Model 2	
Rho-squared			0.16		0.18	
Final log-likelihood			−149.13		−143.63	
Akaike information criterion			467.26		460.26	
Bayesian nformation riterion			953.26		897.86	

OR, odds ratio; CI, confidence interval; HBM, health belief model. Model 1: all covariates except the demographic factors and health factors were included in the analysis; Model 2: all covariates were included in the analysis.

## Data Availability

Not applicable.

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
