# Peer review of "Cognition and Behavior of COVID-19 Vaccination Based on the Health Belief Model: A Cross-Sectional Study"

_vaccines, 2022, doi:10.3390/vaccines10040544_

Round 1
Reviewer 1 Report
This is an interesting article that is written well for the most part, however there are signifiant updates needed for word choice, repetitive sentences, and run on sentences.
The scientific ideas presented are quite good but there are a few major concerns that I would like to see addressed.
First, there is a lot of discussion on the statistical analyses in section 2.3 but it is unclear how this is really being used in the results. If you are going to talk about AIC and BIC methods, then the results need to be clear as to how they helped identify your hypotheses.
This is listed in the beginning of the text but when they get to the HBM based heterogeneity factors there is limited correlation provided. This is a critical part of the paper and much more effort needs to go into explaining how there methods were incorporated into the data analysis.
This section also discussed the odds ratio with no real explanation. This is also presented in the abstract. These are key points of the work the need to be clearly presented to the reader as to their significance.
Second, the figures need to have more explanation in the caption as to what the reader is looking at. I would like to see the figures stand alone so that the reader does not have to search through the text to see what is being presented.
Third, as stated above, there needs to be a grammatical improvement. Here are some examples but this is not at all inclusive:
Abstract: main sources of COVID-19 cognitive and vaccination...
Page 2, second paragraph: epidemic appears deviation
Last sentence of this paragraph is complete run on
Fourth, some of the percentages in Table 2 do not match with the listing of percentages in the text.
Lastly, the Table 1 S. is listed at the end of the paper with no reference in the text.
In the end, this paper has strong potential but the authors need to highlight the significance better. Also, the statistical methods that are mentioned need to be clearly identified in the analyses.
Reviewer 2 Report
the manuscript covers an interesting topic regarding the acceptability of covid-19 vaccine, the introduction provides sufficient bacgroung information, the methodology is well described and results are clearly presented.
The manuscript aims to explore vaccine cognitions, risk perception and health behavior of COVID-19 in China And the topic original or relevant in the field. It is relevant and original since COVID-19 is still an important health challenge It used a health believe model not previously applied in understanding covid-19 vaccination acceptance I do not have any particular suggestion. The method is appropriate and clearly reported The conclusions consistent with the evidence and arguments presented and do they address the main question posed. The references appropriate.The tables and figures. They are appropriate. A graphical abstract might improve the readability of the manuscript and interest of the readers
Figures and tables are well formatted and easy to read.
lines 79-81 is more a discussion than aim. please move it
please add a brief description of Wenjuanxing
conclusions are coherent with the results
Round 2
Reviewer 1 Report
Paper just needs a bit more editing with spelling and grammar.